# Analysis on the Influence of Incident Light Angle on the Spatial Aberrations of Acousto-Optical Tunable Filter Imaging

**DOI:** 10.3390/ma15134464

**Published:** 2022-06-24

**Authors:** Kai Yu, Huijie Zhao

**Affiliations:** 1School of Instrumentation Science & Opto-Electronics Engineering, Beihang University, No. 37 Xueyuan Road, Haidian District, Beijing 100191, China; 2Key Laboratory of “Precision Opto-Mechatronics Technology”, Ministry of Education, No. 37 Xueyuan Road, Haidian District, Beijing 100191, China; 3Institute of Artificial intelligence, Beihang University, No. 37 Xueyuan Road, Haidian District, Beijing 100191, China

**Keywords:** AOTF spectrometer, aberrations, coordinate mapping

## Abstract

Acousto-optical tunable filter (AOTF) does not conform to the pinhole model due to the acousto-optic interaction. A calculation method of AOTF aberrations under the condition of incident light with a large arbitrary angle is proposed to solve the problem of coordinate mapping between object space and image space of the AOTF system without refractive index approximation. This approach can provide accurate pointing information of the interested targets for the tracking and searching system based on AOTF. In addition, the effect of cut angle values of the paratellurite crystal on aberrations was analyzed to optimize the design of AOTF cutting according to different application requirements. Finally, distribution characteristics and quantitative calculation results of AOTF aberrations were verified by experiments with different targets, respectively. The experimental results are in good agreement with the simulations.

## 1. Introduction

Photoelectric tracking and searching system is the most important way of passive detection at present. With the development of infrared technology, all-day and all-weather detection can be realized. At the same time, more advanced jamming technologies have emerged accordingly. These jamming techniques can make the target and the decoy highly similar in geometry, kinematics, and radiance within a wide spectral range. The consequence is the failure of traditional detection and identification methods. Therefore, it is necessary to introduce the characteristics of the spectral dimension. For example, the apparent temperature, the gradient of temperature, or the emissivity of the target can be obtained by multispectral detection [1,2,3]. The features of these kinds are very stable and different from the bait, so they can be used to distinguish the target from the bait. Thus, spectral detection plays an important role in anti-jamming imaging systems now and in the future.

Spectral imaging detection includes optical system design, mechanical design, photoelectric conversion technology, target detection, recognition algorithm, and many other fields. However, to adapt to the rapid changes in the scene, there is an urgent need for an imaging method with a flexible spectrum. The acousto-optic tunable filter (AOTF) is a kind of narrowband filter based on birefringent crystal material. The wavelength, direction, and intensity of diffracted light can be modified by modulating the sound field with a high-frequency sinusoidal signal. Moreover, it has many advantages, such as being electronically controlled with high speed, wide spectral range, and all-solid-state. As a result, AOTF staring imaging spectrometer is commonly used in spectral analysis [4], laser shaping [5,6], spectral microscopic [7,8], and many other fields [9,10,11,12,13]. Similarly, the advantages of AOTF make it very suitable for tracking and searching system.

As we know, AOTF is mostly applied to the spectral imaging detection of static targets due to the different application requirements before. In addition, the field angle of view of the AOTF system is generally designed to be small (approximately equal to ±2°) because of the constraint of the separation angle. Additionally, it is considered that the imaging aberration caused by AOTF is very small and can be ignored. Therefore, within a small angle range, many scholars only considered the impact of incident light angle on device performance. Zhang Zhonghua analyzed the influence of the incident light angle on the spectral bandwidth, diffraction efficiency, and the separation angle inside and outside the crystal; however, their analysis is limited to the interaction plane of AOTF [14,15,16,17]. Yushkov gave the analytical expressions of the central wavelength and the tuning frequency under oblique incidence [18], but for some special applications, the precise geometric information of the target is required. At this time, the aberration caused by AOTF can not be ignored even under the condition of a small field angle of view. Pozhar and Machikhin focused on the quantitative description of AOTF spatial aberrations. In their previous works, an approximation of small birefringence was made, and they only analyzed the aberrations in the direction of the optical axis and the direction perpendicular to the optical axis. Then they extended the analysis of AOTF spatial aberrations from two-dimensional space to three-dimensional space [19,20,21,22,23] and proposed a method of correcting spectral and spatial aberrations by AOTF pre-calibration for applications in which geometric characteristics shall be attained at the same time, such as microimaging [24]. Later, they used the developed DLL to combine the ray-tracing model with ZEMAX and applied it to the optical design of the stereoscopic spectral imaging system [25,26,27]. In addition, they used a single AOTF to obtain the three-dimensional information while obtaining the spectral information of the target [28]. Nonetheless, no quantitative experimental results of AOTF aberrations were given. In a word, the influence of incident light angle on AOTF can be summarized into several stages. From the analysis within the two-dimensional plane (acousto-optic interaction plane) to three-dimensional space, from the approximate analytical solution neglecting the change in refractive index to the accurate iterative solution. Until now, the acousto-optic interaction model based on momentum matching conditions has been relatively perfect.

However, if AOTF is used for tracking and searching systems, different issues need to be considered. The number of pixels occupied by the detected target on the image plane is relatively few, and the angle of light entering AOTF may no longer be constrained by the separation angle because there is no overlap between the zero-order light imaging and the diffracted light imaging. In other words, the zero-order light and the first-order diffracted light are allowed to be imaged at the same time under the condition that the imaging is not saturated. Moreover, zero-order light imaging can be eliminated through image processing because it always exists stably, and the intensity of diffracted light fluctuates with the change in driving frequency. Therefore, for AOTF used for tracking and searching system, there are two main problems to be solved: (1) How can the problem of central wavelength drift, that is, to ensure the accuracy of the spectral data of the target, be solved? (2) How can the accurate pointing information (azimuth and pitch angle) of the target be obtained? As photoelectric detection equipment, it is one of the most basic functions to give the target pointing information accurately. Otherwise, additional optoelectronic equipment is required for assistance.

As the field of view increases, the drift of the central wavelength of AOTF becomes more serious. Some scholars corrected the drift through the computational technique to achieve the purpose of expanding the field of view [29,30]. It means that problem (1) mentioned above can be solved for the AOTF system with a large field of view. On the other hand, the expansion of the field of view also means that the aberration of AOTF is more severe. The AOTF cannot use the pinhole model in ray tracing. We first conducted an in-depth study of AOTF aberration to solve the problem (2). If the calculation and analysis of aberration of AOTF can ensure sufficient accuracy, AOTF can be extended to the tracking and searching system of moving targets. Therefore, it is necessary to analyze the influence of different incident angles on imaging aberrations in a large range of angles. As far as we know, the study of this kind is not complete enough and lacks experimental verification. In this article, we analyzed the influence of the angle of incident light on AOTF spatial aberrations in three-dimensional space by the ray-tracing method and show quantitative results of the experiments. Moreover, the influence of cut angle values of the paratellurite crystal on aberrations was simulated.

The main contribution of this paper was to deeply analyze the aberration of AOTF with a large field of view so that it can give the target pointing information accurately when AOTF is used for the detection of moving targets and put forward a new application direction of AOTF. For the AOTF imaging spectrometer with a small field of view, the aberration is very small. AOTF can be considered to conform to the pinhole model, and the method in this paper is not required for ray tracing. In addition, the algorithm needs to be optimized in the real-time system due to the use of the iterative algorithm.

## 2. Method

At present, the AOTF imaging spectrometer can be classified into two types of structures according to the light entering the AOTF. One is with parallel entering light, and the other is with the convergent entering light. Both have their advantages; in theory, the lateral chromatic aberration of the confocal structure is less, while the axial chromatic aberration of the collimated optical path is less. Considering that lateral chromatic aberration can be corrected by the wedge angle of the AOTF rear surface (the rear wedge angle of AOTF we used is 4.83°), so the collimated light structure is used for the simulations and the experimental design.

The analysis of aberration is essential to analyze the relationship between the incident light angle of AOTF and the exit direction of diffracted light. As a spectrometer, the ray-tracing method can no longer be limited to the crystal but needs to be converted to the world coordinate system for analysis. The advantage of this method is that it has better adaptability. It can still analyze and guide the design of the optical system when changing the cut angle values of AOTF or the angle of Bragg’s diffraction.

### 2.1. Ray Tracing of Arbitrary Light in Three-Dimensional Space

In the design and performance analysis of AOTF devices, monochromatic light is usually employed as the light source. It should be pointed out that in many analyses of AOTF device modeling, the incident light is usually vertical, and a certain diffracted light is selected by changing the driving frequency that matches the wavelength of the incident light. Momentum mismatch occurs when the incident light is not perpendicular to the AOTF front surface, or the wavelength of the incident light does not match the driving frequency. However, in actual applications of spectral imaging detection, the incident light is mostly polychromatic, and the diffracted light also essentially satisfies the momentum matching condition in another wave vector diagram in which the wavelength of incident light drifts.

Take e-polarized incident light and o-polarized diffracted light as an example. When the incident light is white, the driving frequency is fixed, and the angle of the incident light is changed, then the original incident wavelength $\left(\lambda_{1}\right)$ no longer satisfies the momentum matching condition, as the blue triangle shown in Figure 1a. Additionally, there is another wavelength $\left(\lambda_{2}\right)$ of incident light and the sound vector to meet the new momentum matching condition, such as the red triangle shown in Figure 1b, the wavelength and direction of the diffracted light can be calculated accordingly. In Figure 1a,b, the length and direction of $L_{AB}$ and $L_{DE}$ do not change, which means the sound vector remains unchanged. In other words, when the incident light is polychromatic, the diffracted light of a certain wavelength can always be found to meet the momentum matching condition. The momentum mismatch is only an artificially defined quantity to calculate the diffraction efficiency or spectral bandwidth. If the intensity of the sound field is constant, the amount of momentum mismatch is only related to the direction and wavelength of the diffracted light. Momentum mismatch is not considered in the analysis of AOTF imaging aberrations. Therefore we need to find the wavelength and exit direction of diffracted light that meet the conditions.

In the world coordinate system, assuming the direction cosine of the incident light is $\left(L_{0},M_{0},N_{0}\right)$, in the crystal coordinate system, the direction cosine of the incident light is converted to $\left(L_{1},M_{1}\cos\theta_{i}^{*}-N_{1}\sin\theta_{i}^{*},M_{1}\sin\theta_{i}^{*}+N_{1}\cos\theta_{i}^{*}\right)$ and $\theta_{i}^{*}$ represents the angle between the front surface of AOTF and the 110 axes. The first surface [31] adopts the refraction theorem and the equation for calculating the refractive index of extraordinary light, as shown in Equation (1).
(1)$$\left\{\begin{array}{l} n_{air}\sin\theta_{in}=n_{e}(\theta )\sin(\theta -\theta_{i}{}^{*}),\\ n_{e}^{2}(\theta )=\frac{n_{o}^{2}n_{e}^{2}}{n_{o}^{2}\sin^{2}\theta +n_{e}^{2}\cos^{2}\theta }, \end{array}\right.$$
where $n_{air}$ is the refractive index of the air medium, $\theta_{in}$ is the incident angle in the air medium, $n_{e}(\theta )$ is the refractive index of e light corresponding to the refracted light in the crystal, θ and $\theta_{i}{}^{*}$ is defined as shown in the partially enlarged view of Figure 2, θ is the angle between the refracted light in the crystal and the crystal axis in the polar coordinate system, $\theta -\theta_{i}{}^{*}$ is the angle between the refracted light in the crystal and the normal of the interface, and $n_{o}$ and $n_{e}^{}$ represent the refractive index of o light and e light, respectively. Then, the phase velocity direction of the incident light in the crystal can be obtained. Point A $\left(x_{a},y_{a},z_{a}\right)$ is the intersection point of the incident light, and the ellipsoid of the light wave vector shown in Figure 1. $\left|K_{a}\right|$ can be obtained with driving frequency corresponding to the wavelength of the incident light and the direction of the sound vector $K_{a}$ under the condition of momentum matching when the incident light is perpendicular to the front surface of AOTF. Since the driving frequency has not been tuned and $K_{a}$ remains the same, even if the incident light angle has been changed, the coordinates of point B $\left(x_{a}-k_{\mathrm{ax}},y_{a}-k_{\mathrm{ay}},z_{a}-k_{\mathrm{az}}\right)$ can be calculated from the coordinates of point A. The coordinate of point C $\left(x_{c},y_{c},z_{c}\right)$ is the intersection of the extension line of OB and o-wave vector sphere, and the length of the BC line, $L_{BC}$, can be obtained.

It can be found that the value of $L_{BC}$ is related to the incident wavelength. It can be considered that the incident light vector, diffracted light vector, and sound vector satisfy the momentum matching condition when $L_{BC}$ is close to 0 with the change in incident light wavelength. Therefore, the corresponding central wavelength at an arbitrary incident angle is expressed by Equation (2). The “argmin” in Equation (2) means that the center wavelength corresponds to the smallest $L_{BC}$ when the wavelength changes within a certain range.
(2)$$\lambda_{\mathrm{center}}=\underset{\lambda_{\min}\leq \lambda \leq \lambda_{\max}}{\mathrm{argmin}\;(L_{BC}(\lambda ))}.$$

The intersection point of the incident light and the E-light ellipsoid can be calculated, i.e., D $\left(x_{d},y_{d},z_{d}\right)$, after we obtain $\lambda_{\mathrm{center}}$. It can be understood that in the process of wavelength change under the condition of a certain angle of the incident light, point A moves to point D. Thus far, the intersection point E $\left(x_{e},y_{e},z_{e}\right)$ of the diffracted light and the o-light sphere, and the exit direction of diffracted light in the crystal coordinate system can be obtained simultaneously. After the diffracted light passes through the AOTF exit surface and then changes from the crystal coordinate system to the world coordinate system, the directional cosine of the diffracted light($\vec{GH}$ in Figure 2) can be obtained. The iterative method must be used to obtain more accurate ray tracing because the refractive index of tellurium dioxide changes with the wavelength, and the analytical solution is no longer applicable.

Through this method, the exit direction of diffracted light corresponding to incident polychromatic light at any angle can be calculated, and further its imaging position, then the corresponding relationship between AOTF spatial aberrations and the angle of incident light can be calculated. This method is also applicable to the cases where the incident light is o-polarized, the diffracted light is e-polarized, and cases about the different cut angle values of the paratellurite crystal.

### 2.2. Aberration Simulation of AOTF at Different Incident Angles

Through the method above, it can be known that another wavelength of the incident satisfies the momentum matching condition with the original sound vector when the white light is incident obliquely. Select the incident light wavelength λ = 632 nm, driving frequency f = 73.25 MHz, and calculate the matching center wavelength under the condition of oblique incidence within the range of ±4.0° polar angle and azimuth angle. The ultrasonic cut angle of AOTF is 6.5°. The rear wedge angle is 4.83°. These parameter settings are consistent with the actual design parameters of the AOTF cutting.

As shown in Figure 3, the central wavelength drifts with the increase in the incident light angle. In the azimuth plane, the drift of the central wavelength is symmetrically distributed. In the polar angle plane, the drift of the central wavelength is asymmetric, and the drift of the central wavelength near the AOTF crystal axis is greater than that away from the crystal axis when the incident light is inclined by the same angle from the vertical incidence to both sides.

The direction of diffracted light can be further calculated and compared with the image plane position of diffracted light under the condition of pinhole imaging after the central wavelength is obtained, and the aberrations of AOTF can be obtained accordingly. The focal length of the rear objective lens is 50 mm, and the pixel size is 4.5 μm.

The $L_{\left(AC\right)}$ in Figure 4c corresponds to the ordinate in Figure 4b, and the $L_{\left(BC\right)}$ in Figure 4c corresponds to the ordinate in Figure 4a. AOTF aberrations are different in polar direction and azimuth direction significantly. In the azimuth direction, the aberration is 180° rotational symmetry, and the aberration is less than two pixels within a ±4° incidence angle. In the polar direction, the aberration follows the symmetrical distribution of the azimuth plane, and the aberration caused by oblique incidence is considerable. The larger the tilt angle is, the greater the aberration is, up to 30 pixels within ±4° incidence angle. It can be discovered from the comparison between Figure 3 and Figure 4 that the greater the center wavelength drifts, the more serious the AOTF aberration is.

The accuracy of the simulation mainly considers the difference between the design value and the theoretical value of the ultrasonic cut angle.

As shown in Figure 5, the error of edge field aberration does not exceed two pixels when the error of ultrasonic cut angle is ±0.2°. The aberration error in the acousto-optic action plane is mainly considered here because the error in the azimuth plane affected by the ultrasonic cut angle is very small, so it is not shown. In the simulation, the principle of parallel tangent is no longer satisfied due to the error of the ultrasonic cut angle.

In addition, cut angle values of the paratellurite crystal, i.e., α, affect the aberrations of AOTF. The aberrations in the polar direction of AOTF with different α are simulated, as shown in Figure 6. The results show that the aberrations of AOTF decrease with the increase in α within the range of (0°,18.9°).

## 3. Results

### 3.1. Experiment of the Aberration Distribution Characteristics with Integrating Sphere as the Target

We chose a small area target, the integrating sphere, as the light source to analyze the distribution characteristics of AOTF distortion more intuitively. The integrating sphere had a diameter of 12 inches. Such target selection can make the diffracted light imaging not overlap with the zero-order light imaging so that the influence of incident angle on aberrations of AOTF can be analyzed in a wider range.

The experimental imaging setup of different incident angles of the integrating sphere is shown in Figure 7. φ means the half-angle aperture in the azimuth direction (plane perpendicular to the acousto-optic interaction plane). The distance between the integrating sphere and AOTF was 2.8 m, the outlet diameter of the integrating sphere was 4 inches, the aperture of AOTF was 20 mm, the ultrasonic cut angle was 6.5°, the rear wedge angle was 4.83°, the spectral range was 400–1000 nm, the change in polarization was e→o, the focal length of the rear objective lens was 50 mm, and the pixel size was 4.5 μm. AOTF was fixed on the precision turntable, the driving frequency of the AOTF was adjusted by the PC, and the camera could obtain images of diffracted light with different wavelengths. Images of diffracted light at different incident angles can be obtained through the precision turntable.

The distance between the target and AOTF is far in the experimental setting so that the pre-collimating mirror group can be omitted. It is only necessary to measure the aberration of the imaging mirror group and add the aberration parameters to the simulation model.

In Figure 8, the diameter of the large dots is 15 mm, the smaller is 8 mm, and the distance between the calibration points is 40 mm. The coordinates of the calibration point were obtained by the method of geometric center extraction, and the unsatisfactory calibration points were eliminated. Finally, the radial distortion coefficient of the imaging lens group could be obtained by Zhang Zhengyou’s calibration method, and the result is shown below as Equation (3).
(3)$$\left(\begin{array}{c} k_{1}\\ k_{2}\\ k_{3} \end{array}\right)=\left(\begin{array}{c} 0.37667\\ -3.40922\\ -0.00195 \end{array}\right).$$

As shown in Figure 9, the angle of incident light from left to right gradually approaches the AOTF crystal axis, and the image of diffracted light of the integrating sphere has changed. In general, the aberrations present an asymmetric distribution in the polar direction. By taking the image of polar angle=0° (vertical incidence) as the benchmark, the imaging changes are small when the incident polar angle gradually increases in the direction away from the crystal axis; thus, the spatial aberrations of the AOTF in this direction are small. The imaging of the integrating sphere changed significantly when the incident polar angle increased in the direction close to the crystal axis; thus, the spatial aberrations of the AOTF in this direction are tremendous.

The influence of incident azimuth angles on the aberrations of AOTF can be obtained with the same experimental method by rotating the AOTF by 90° around the optical axis. From Figure 10, the aberrations caused by the change in the azimuth incidence angle still exist mainly in the polar direction and are distributed symmetrically up and down based on the image of azimuth angle=0°.

Therefore, we can conclude that the aberrations of AOTF mainly exist in the polar direction, the amount of aberrations in the azimuth direction is small, the aberrations are affected by the incident angle in the polar direction greatly, and the closer the incident angle to the AOTF crystal axis is, the greater the aberrations are.

### 3.2. Quantitative Experiment with the Square Grid as the Target

We chose to quantitatively analyze the influence of incident angle on AOTF aberrations in the polar direction. The experimental target is vertically placed at a distance of 5 m from AOTF, and other experimental conditions are consistent with those in Section 3.1. The imaging of the target center point, $\left(x_{55},y_{55}\right)$ in Figure 11a, is located at the center of the image plane. The size of the target is 350 mm × 350 mm, the length of the small square grid is 9 mm, and the distance between adjacent squares is 40 mm.

In Figure 11b, the blue star points represent the ideal points of pinhole imaging, the solid green dots represent the simulated imaging points, and the red boxes indicate the actual imaging areas of the square grid. The coordinate of the actual image point is obtained by extracting the geometric center of the sampling points. We found that the simulated imaging points were located near the actual imaging areas. The reason for the errors may include the surface of the target is not strictly perpendicular to AOTF, there exists an error in distance measurement and target center extraction, and the sound field of AOTF is not ideal. The $error_{RMS}$ can be calculated by Equation (4).
(4)$$error_{RMS}=\sqrt{\frac{1}{81}\sum_{i=1}^{9}\sum_{j=1}^{9}\left[{\left(x_{actual,ij}-x_{model,ij}\right)}^{2}+{\left(y_{actual,ij}-y_{model,ij}\right)}^{2}\right]}.$$
where $x_{actual,ij}$ is the row coordinate of the diffracted light imaging position. $x_{model,ij}$ is the row coordinate of the calculated imaging position. y is the column coordinate. The $error_{RMS}$ between the actual imaging points and the ideal imaging points is 39.70≈40 pixels, and the deviation mainly exists in the polar direction. The $error_{RMS}$ between the actual imaging points and the modeled imaging points is 2.14≈2 pixels, in which the coefficient of fixing errors is fitted by the least square method. The pixel difference of each point is shown in Figure 12. The sequence of the labels of the sampling points is to take the upper left point as the first sampling point. The sampling is in the row direction.

As can be seen from Figure 12a, the pixel difference of all 81 sampling points between the actual target position and the simulated target position is less than 3.5 pixels. In Figure 12b, the closer the sampling point is to the right of the image, the smaller the aberration is, which is consistent with the previous analysis.

We made the polar angle closer to the AOTF crystal axis, and the result is shown in Figure 13.

By comparing Figure 11b and Figure 13b, it was found that the aberrations of AOTF became more severe with the increase in polar angle, and the black squares on the left were deformed, which is consistent with the results of the previous analysis and simulations.

## 4. Conclusions and Future Directions

The angle of incident light influences the spatial aberrations of AOTF imaging, and the aberrations mainly exist in the direction of the polar angle. In addition, the influence of cut angle values of the paratellurite crystal on aberrations was analyzed. In the range (0°,18.9°), the larger the ultrasonic cut angle is, the smaller the spatial aberrations of AOTF imaging are.

In this article, we verified the simulation results through quantitative experiments. The results show that the $error_{RMS}$ between the simulation results and the experimental results is about two pixels when the incident angle in the polar direction is −5.3° under the experimental parameter settings above. Moreover, it provides some guidance for the design of the AOTF system with a large field of view. The computational accuracy of AOTF aberration guarantees that AOTF has the potential to be used in tracking and searching system, which is still blank in the previous research.

Compared with other spectroscopic methods, the biggest advantage of AOTF lies in the staring spectral imaging. Therefore, AOTF can be used not only for the spectral detection of static targets but also for the detection of moving targets, which can expand the application range of AOTF in the future.

## Figures and Tables

**Figure 1 materials-15-04464-f001:**
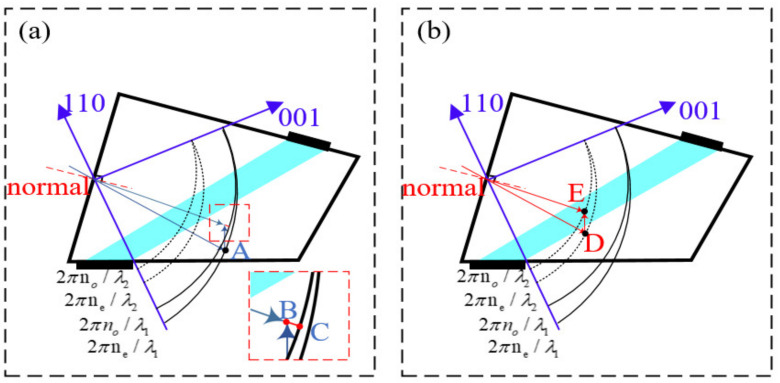
Wave diagram of anisotropic AO diffraction. Momentum mismatching (**a**) and momentum matching (**b**) under oblique incidence.

**Figure 2 materials-15-04464-f002:**
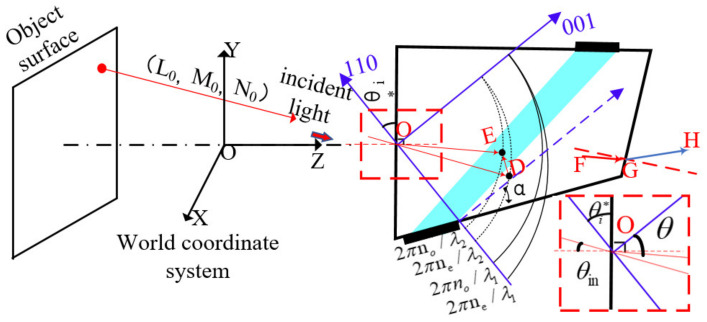
Ray tracing of AOTF in three-dimensional space.

**Figure 3 materials-15-04464-f003:**
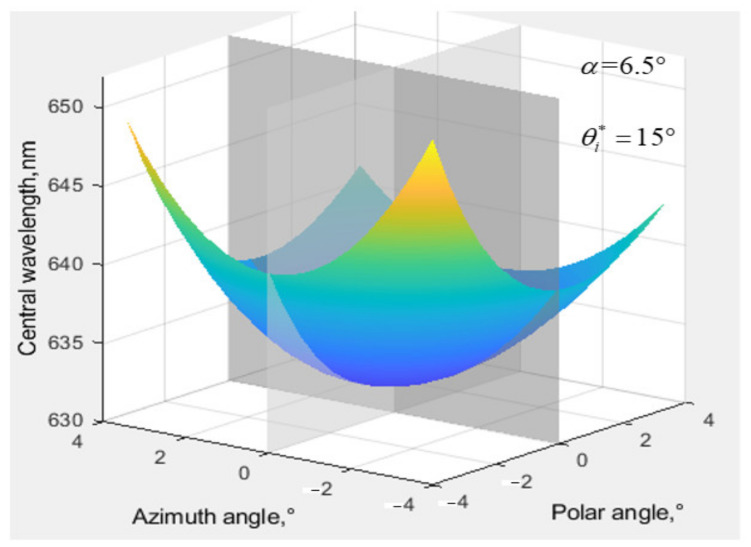
Central wavelength shift at ±4° angle of incident light.

**Figure 4 materials-15-04464-f004:**
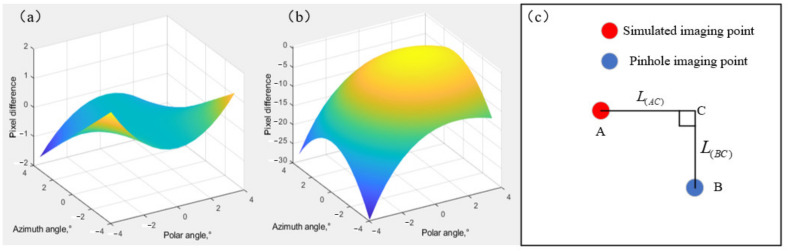
The difference between the imaging position of diffracted light at different incident angles of AOTF and pinhole imaging: (**a**) the difference in azimuth direction and (**b**) in polar direction, and (**c**) the image plane.

**Figure 5 materials-15-04464-f005:**
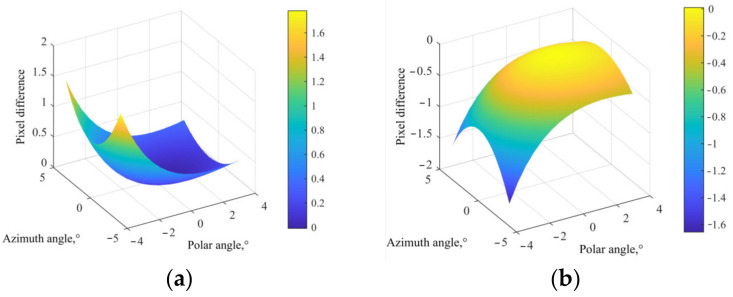
The accuracy of the simulation (**a**) α = 6.3° (**b**) α = 6.7°.

**Figure 6 materials-15-04464-f006:**
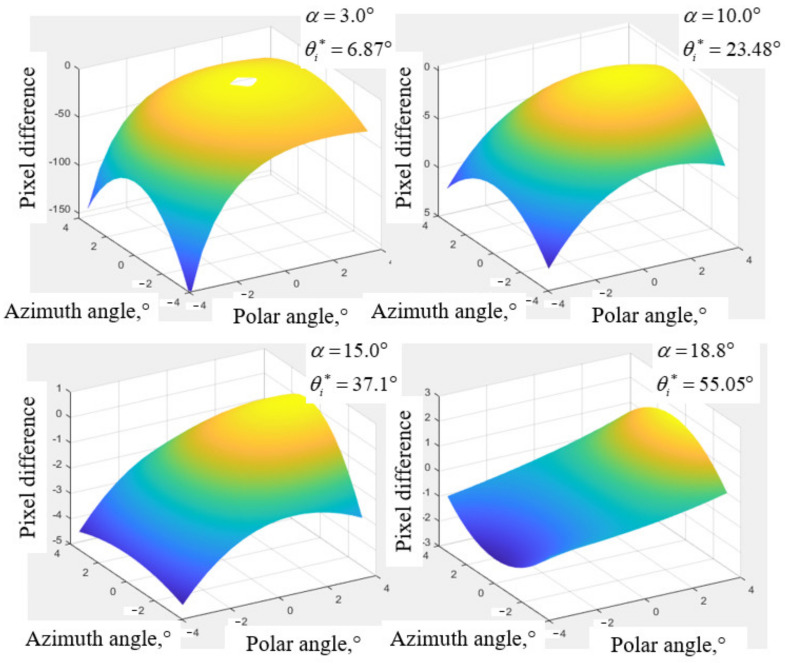
Aberrations of different ultrasonic cut angles of AOTF.

**Figure 7 materials-15-04464-f007:**
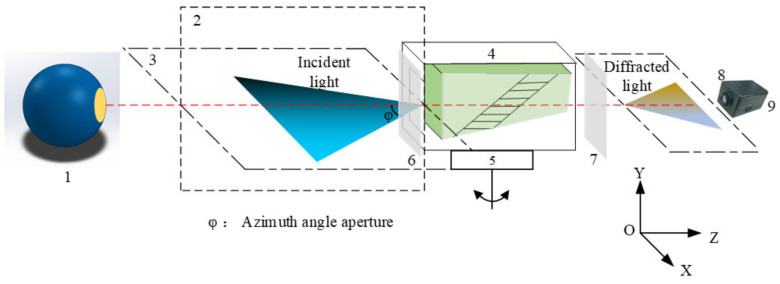
Integrating experimental sphere setup of AOTF based on different incident light angles: 1—integrating sphere; 2—polar plane; 3—azimuth plane; 4—AOTF; 5—rotatable stage; 6 and 7—polarizers; 8—imaging lens; 9—camera.

**Figure 8 materials-15-04464-f008:**
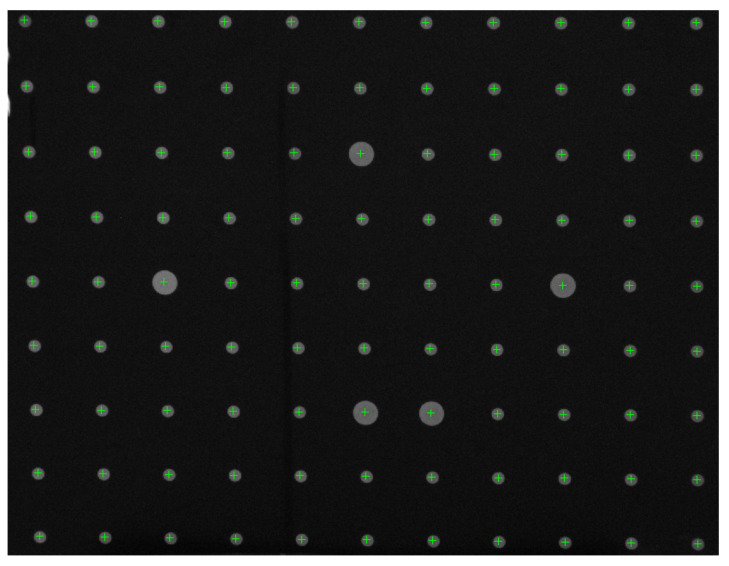
Distortion test of imaging lens group.

**Figure 9 materials-15-04464-f009:**
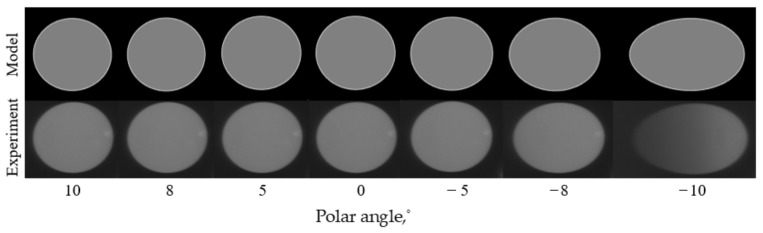
Simulated (upper row) and experimental (lower row) images with the AOTF tuned to λ = 632.8 nm (f = 72.84 MHz) for different angles of incident light in the polar plane.

**Figure 10 materials-15-04464-f010:**
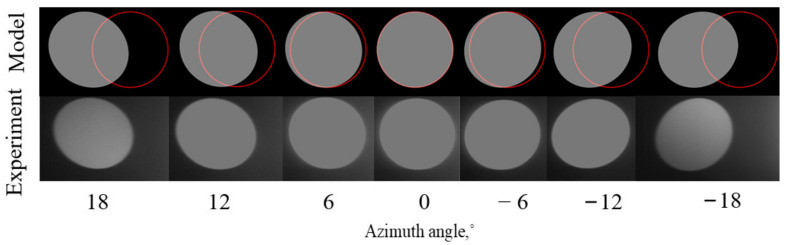
Simulated (upper row) and experimental (lower row) images with the AOTF tuned to λ = 632.8 nm (f = 72.84 MHz) for different angles of incident light in the azimuth plane. The solid red line in the upper row indicates the ideal positions of the integrating sphere with the assumption of pinhole imaging.

**Figure 11 materials-15-04464-f011:**
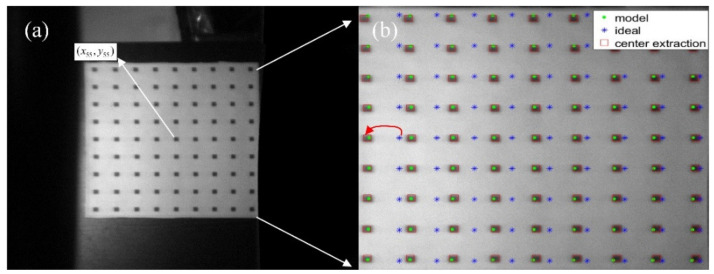
The experiment of the square target: (**a**) actual image of diffracted light at a polar angle of −5.3°; (**b**) the partially enlarged view of the square target.

**Figure 12 materials-15-04464-f012:**
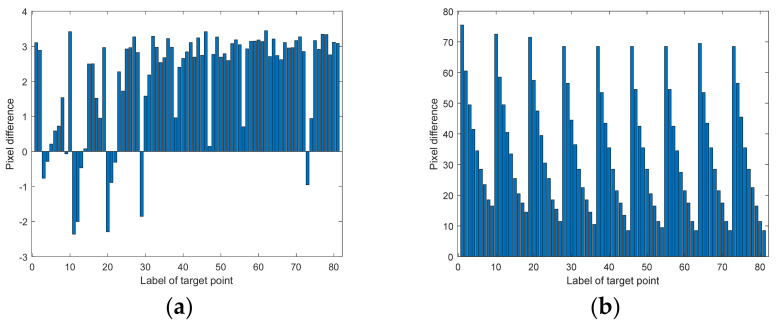
The pixel difference of each point: (**a**) pixel difference between the actual target position and simulated target position; (**b**) pixel difference between the ideal target position and actual target position.

**Figure 13 materials-15-04464-f013:**
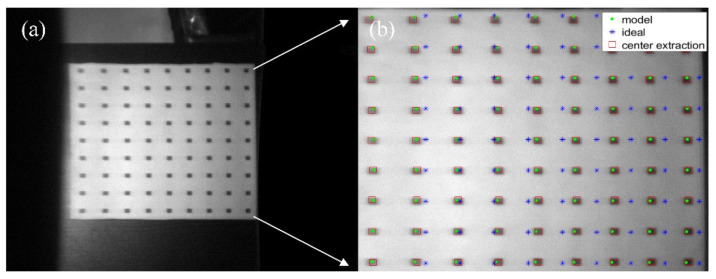
The experiment of the square target: (**a**) actual image of diffracted light at a polar angle of −7°; (**b**) the partially enlarged view of the square target.

## Data Availability

Not applicable.

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
