# Peer review of "Analysis on the Influence of Incident Light Angle on the Spatial Aberrations of Acousto-Optical Tunable Filter Imaging"

_materials, 2022, doi:10.3390/ma15134464_

Round 1
Reviewer 1 Report
The authors of this article discuss a method for calculating the aberrations of an acousto-optic tunable filter (AOTF) under conditions of incident light with an arbitrary angle.The influence of the incident angle on the AOTF aberrations is analyzed in three-dimensional space without approximation of the refractive index at the system level, and not at the device level. The authors claim that this approach can make target depth information more accurate when AOTF is used to obtain 3D information, or be useful for geometric calibration of an AOTF spectrometer.
In addition, the influence of the values of the paratellurite crystal cutoff angle on aberrations is analyzed. Finally, the distribution characteristics and the results of quantitative calculation of AOTF aberrations are verified by experiments with different targets, respectively.
In general, the article deserves to be published in the journal Materials, but it is necessary to slightly improve the presentation.
First of all, in the title of the article, the abbreviation AOTF should be replaced with an acousto-optical tunable filter, and the same should be done in the Abstract, where is this abbreviation first mentioned.
Secondly, the first and second lines of Eq.(1) should end with a comma, while the next sentence should start with “where ”.
Thirdly, equation 2 must end with a dot and must be deciphered "argmin"!
Fourth, what does φ mean in Fig.6!
Fifth, the sentence in line 248 should end with the words “the result is shown below as”, and equation 3 should end with a dot.
Sixth, the sentence in line 293 should end with the words “can be calculated by equation 4”, and equation 4 should end with a dot.

Reviewer 2 Report
The entire paper should be planned, arranged, and written again. The paper in its current form is difficult to understand.
A calculation method of AOTF aberrations under the condition of incident light with an arbitrary angle is proposed. The effect of the incident angle on the AOTF aberrations is analyzed in three-dimensional space without refractive index approximation at the system level instead of the device level. This approach may make the depth information of the target more precise when AOTF is used for 3D information acquisition, or be useful for geometric calibration of AOTF spectrometer. In addition, the effect of cut angle values of the paratellurite crystal on aberrations is analyzed. Finally, distribution characteristics and quantitative calculation results of AOTF aberrations are verified by experiments with different targets respectively. The experimental results are in good agreement with the simulations.
The purpose of the study, deliverables, and utilities are not clearly explained. Generally, sentences do not start with To,,,,, but there are many such instances.
The novelty of the study is not at all clear.
Simulation and experimental setup not explained or justified.
Research methodology is to be justified.
Accuracy of the experiment/simulation not highlighted.
Challenges to the current approaches for calculation of AOTF aberration are not documented well.
The paper would need significant time to improve. If not possible, it can be rejected.
Reviewer 3 Report
The paper has enough contribution. The following points can improve the manuscript.
- In the Abstract, change this “Acousto-optical tunable filter è Acousto-optical tunable filter (AOTF).”
- Enhance the introduction to show the main contribution of this work.
- Please double-check all equations to be true. (compulsory)
- There should be some discussions on the limitations of the proposed approach.
- References should be updated. There is only one reference in 2021.
- Change the “Conclusion” section title to “conclusion and future directions” and add future directions to the research. (Better to be changed).
- Enhance the English of the work using at least Grammarly. There are some problems with paper typesetting.
- The paper is suitable for acceptance but needs to address the comments mentioned above.
Round 2
Reviewer 2 Report
I am still not satisfied with your responses.
You are doing "Analysis on the influence of incident light angle on the spatial aberrations of acousto-optical tunable filter imaging".
The purpose of your analysis is not explained well. How it is going to help the industry. Usefulness is not explained well.
You need to highlight in simple English the weakness of the present-day imaging system. Then write the research gap/ research problem. How/why the influence of incident light angle on the spatial aberrations plays a very important role---Justify. How previous researchers attempted to address this research problem. How/why your effort is novel. Justify your methodology used. Efficiency/effectiveness of your approach. Statistical verification of the result. How your technique can be scaled up to an industrial scale with the development of imaging system/camera etc.
I recommend rewriting the paper using simple correct English, taking all our input.
